# Analysis of SARS-CoV-2 Spike Protein Variants with Recombinant Reporter Viruses Created from a Bacmid System

**DOI:** 10.3390/ijms24098156

**Published:** 2023-05-02

**Authors:** Arne Cordsmeier, Doris Jungnickl, Alexandra Herrmann, Klaus Korn, Armin Ensser

**Affiliations:** Institute for Clinical and Molecular Virology, Friedrich-Alexander-Universität Erlangen-Nürnberg, 91054 Erlangen, Germany; arne.cordsmeier@uk-erlangen.de (A.C.); klaus.korn@uk-erlangen.de (K.K.)

**Keywords:** SARS-CoV-2, variants of concern, Spike protein

## Abstract

SARS-CoV-2, the causative agent of COVID-19, has spread around the world with more than 700 million cases and 6.8 million deaths. Various variants of concern (VoC) have emerged due to mutations and recombination and concurrent selection for increased viral fitness and immune evasion. The viral protein that primarily determines the pathogenicity, infectivity, and transmissibility is the Spike protein. To analyze the specific impact of variant Spike proteins on infection dynamics, we constructed SARS-CoV-2 with a uniform B.1 backbone but with alternative Spike proteins. In addition, ORF6 was replaced by EYFP as a biological safety measure, and for use of this well-established reporter. We show that namely the delta variant Spike proteins cause a distinct phenotype from the wild type (B.1, D614G) and other variants of concern. Furthermore, we demonstrate that the omicron BA.1 Spike results in lower viral loads and a less efficient spread in vitro. Finally, we utilized viruses with the two different reporters EYFP and mCherry to establish a competitive growth assay, demonstrating that most but not all Spike variant viruses were able to outcompete wild type SARS-CoV-2 B.1.

## 1. Introduction

SARS-CoV-2 emerged in the end of 2019 in Wuhan, PR China [1] and rapidly spread around the world with, to date, more than 700 million confirmed cases with more than 6.8 million deaths (World Health Organization [2], accessed on 1 February 2023). Like other betacoronaviruses (e.g., SARS-CoV-1, MERS), SARS-CoV-2 comprises a positive-strand RNA genome with a length of 30.000 nt, assembled in a nucleocapsid within a lipid envelope [3,4]. The genome is replicated by an RNA-dependent RNA-polymerase (RdRp) with a proof-reading function [5,6]. Nevertheless, coronaviruses have a mutation rate of 10^−5^ to 10^−3^ substitutions per base and infected cell [7], due to the possible contribution of ABOBEC-related deamination [8,9,10]. For SARS-CoV-2, the mutation rate was determined to be 6.677 × 10^−4^ substitution per site and year utilizing sequences from the USA [11]. 

SARS-CoV-2 infects host cells by binding to the host cell receptor ACE2 (angiotensin converting enzyme 2) via its 1273 amino acids comprising Spike protein. Thereafter, the Spike protein is cleaved by the host cell protease TMPRSS2 transmembrane serine protease 2) for activation and subsequent membrane fusion [12,13]. An alternative entry route comprises the endosomal uptake of the virus after binding to ACE2 (reviewed in [14]). The endosomal entry pathway is for example favored by the omicron variant [15].

The interplay between ACE2 and the SARS-CoV-2 Spike protein is critical for the infectivity of the virus, and for this reason the Spike protein is the major target of neutralizing antibodies. Thus, mutations in this protein are quite common due to immunogenic selection pressure and their influence on infectivity. Numerous mutations have occurred altering infectivity, immune evasion, and transmissibility (e.g., reviewed in [16]), leading to the emergence of variants of concern (VoC) driving the consecutive outbreaks. Furthermore, mutations can be connected to severity of infection and general pathogenic capacity (e.g., reviewed in [17]). Recently, the omicron variants of concern starting with BA.1, that contained more than 30 mutations in the Spike protein, became dominant in the human population from December 2021 onwards (e.g., reviewed in [18]) and account for >90% of infections during the pandemic.

Due to the high number of mutations and their structural interconnection, it is often difficult to assess the impact of specific mutations on infectivity, transmissibility, and immune evasion. Viral isolates do not allow us to deduce the influence of specific Spike mutations, since they may contain multiple mutations in further proteins affecting the replicative properties. Therefore, we developed a convenient, genetically stable bacmid system that allows us to replace every part of the SARS-CoV-2 genome based on homologous recombination [19]. It allows us to directly amplify Spike genes from patient material, also from widely used inactivating sample buffers. Furthermore, it does not depend on virus isolates that are prone to acquire additional mutations in tissue cultures [20,21]. The bacmid is then easily used to recover replicating virus [19]. Here, we utilized this system to assess the impact of different Spike variants within the same backbone. We generated viruses containing VoC Spike genes as well as more exotic Spike genes from viruses causing small local transmission events in the area of Erlangen (Bavaria, Germany). Subsequently, we analyzed their infection parameters, confirming existing knowledge and getting new insights into infection dynamics based on the Spike protein. 

## 2. Results

Here, we utilized the clone B.1 bacmid and homologous recombination [19] to construct SARS-CoV-2 strains containing specific Spike genes (mainly variants of concern) that share the same backbone. These tools allow the assessment of the impact of the different Spike proteins on viral infection and replication in vitro. Homologous recombination was used to insert a kanamycin resistance cassette flanked by unique restriction sites and to remove the original B.1 Spike sequence (D614G). In the second step, this cassette was replaced by the Spike of interest by restriction enzyme digestion and Gibson assembly. Furthermore, all viruses express EYFP as a reporter instead of the ORF6 protein (Figure 1).

Correct cloning and assembly of SARS-CoV-2 Spike variants were confirmed by restriction fragment analysis and by next generation sequencing of the bacmid. All sequences contain the dedicated mutations in the Spike protein (Figure 2). Of note, our wild type B.1, in contrast to the original Wuhan strain, already contains the D614G mutant present in nearly all known variants [22,23]. Next generation sequencing confirmed that the alpha-like variant (B.1.1.7) constructed, further called alpha, additionally contains the N501Y mutation but not the other alpha-typical mutants (e.g., the deletion of 69/70, A570D, P681H), due to the lack of suitable patient material. The beta- and gamma-like variants (B.1.351 and B.1.1.28.1, referred to as beta and gamma, respectively) contain the typical mutations (like E484K and K417N). We analyzed three distinct delta-like variants (B.1.617.2, further called delta, delta49, and delta70) that differ at some positions, while having the same delta-backbone. However, an important difference is seen between delta49 versus delta and delta70, respectively. Delta49 has the characteristic L452R but lacks the T478K, N501Y, and P681R mutations. This mutation pattern is very similar to a delta-like variant detected in Jordan [24]. The Belgian variant (B.1.214.2, further referred to as bel) caused a small outbreak in Franconia. It probably originates from the Democratic Republic of Congo and was brought to Europe by Belgian soldiers, first establishing a restricted outbreak in Belgium [25]. It shows a mutation pattern in its Spike protein that is rather different from the prominent VoC. The omicron-like Spike variant (B.1.1.529) that we analyzed is the BA.1 lineage, containing the typical mutations and an additional R346K mutation.

The phylogenetic relationship of the analyzed variants was compared using bioinformatics tools in the CLC genomics workbench. Therefore, full length viral sequences from patients and the S variants were aligned to each other and phylogenetic trees were constructed (Figure 3). The phylogenetic trees show differences between the distributions of whole genomes sequenced from patients (Figure 3a) and Spike gene sequences only (Figure 3b). For example, in the full genome sequence the Belgian variant showed a larger distance to the other variants while the omicron Spike is most different when only Spike gene sequences are compared.

The construction of recombinant viruses expressing different Spike proteins within the same B.1 backbone enables the focused analysis of their impact on SARS-CoV-2 replication features.

After successful reconstitution of Spike variant containing viruses, we analyzed their replication kinetics using RT-qPCR on viral supernatants (Figure 4). Comparison of the results demonstrated that viruses containing gamma and delta49 Spike tend to replicate faster and reach higher viral loads at 72 h post infection. The delta and delta70 variants, both containing the P681R mutation, showed lower RNA copy numbers (Figure 4a). However, these differences are not significant as visualized by Figure 4b in a logarithmic scale.

To further investigate the impact of Spike variants on the severity of infection, we utilized the reporter viruses to visualize infection phenotypes by fixation of infected cells after 48 h with subsequent fluorescent measurement (Figure 5). The EYFP protein expressed instead of ORF6 shows the morphology of the infected cells (Figure 5a). Here, a homogenous infection could be observed for wild type, alpha, and gamma Spike proteins, while delta and delta70 show massive syncytia formation. The delta49 variant exhibits an intermediate phenotype with moderate syncytia formation. The beta and omicron BA.1 Spike containing viruses depict a weaker extent of infection with lower fluorescent signal. Compared to other variants analyzed, infection with the delta and delta70 variants resulted in a higher number of cell deaths, visualized, and measured by HOECHST staining (Figure 5a,b). The omicron BA.1 Spike containing viruses caused the lowest degree of cell death.

After observing clear differences in cell morphology upon infection with different Spike variants, we asked whether the Spike variants also have an influence on plaque morphology. To this end, CaCo-2 cells were infected with the respective reporter viruses and fixed 48 h post infection. After fixation, plaques were easily visualized due to the EYFP protein that is expressed instead of SARS-CoV-2 ORF6 (Figure 6). Upon analyzing the fluorescent images, the wild type and alpha variant showed the largest plaques while, for the omicron BA.1, only very small plaques were detected (Figure 6a). Measurement of the plaque areas using the ImageJ software (version 1.54d) confirmed the visual observations (Figure 6b). Significant differences are between the plaque sizes of WT and alpha as the largest and omicron BA.1 and beta as the smallest. Furthermore, increased cell death could be observed again for the different delta variants.

Finally, the reporter system was used to perform competitive infection experiments. For this, we employed a recombinant virus expressing mCherry instead of ORF6, containing the B.1 wild type Spike. This was used at equal MOI in co-infection assays with EYFP expressing SARS-CoV-2 containing variant Spikes. Supernatants of these mixed infections were passaged every 48 h and cells were fixed afterwards. The fluorescence signal of the infected cells was measured using the ImageXpress Pico High Content Screening instrument; YFP and mCherry cells were each counted (Figure 7a). The results show that the alpha, delta, delta49, and delta70 variants are able to outcompete the virus containing wild type Spike. In beta, gamma, bel, and, as expected, in the wild type mCherry versus wild type YFP control, no clear advantage for either virus was detected. The omicron BA.1 variant was outcompeted by the wild type (Figure 7b).

## 3. Discussion

The reporter system described in the manuscript is a convenient and powerful tool to analyze specific parts of SARS-CoV-2. Due to the construction of SARS-CoV-2, with the same backbone only differing in the Spike protein, we were able to directly connect the data to the influence of this protein. Furthermore, our system allows the analysis of future variants, in Spike or other proteins, causing restricted outbreaks for their replicative potential by utilizing patient material, avoiding gene synthesis.

An advantage of our system over, e.g., pseudo-typed viral particles, which allow the rapid assessment of infection and by neutralization assays, is the capacity to analyze the impact on replication and plaque morphology. Additionally, our replication-competent viruses can be used in animal experiments, which we plan to do in the future. This will be particularly helpful to address potential pathogenicity-related differences attributable to specific residues in the delta VoC Spike protein. 

Interestingly, in our experiments, the omicron BA.1 Spike variant exhibits the lowest replication capacity on CaCo-2 cells compared to the other variants, although it was the dominant variant in the human population at the time of data collection (Figure 4a,b). The reduced replication of omicron BA.1 in specific cell types has been observed before [26]. An enhanced replication capacity of SARS-CoV-2 omicron in cells of the upper respiratory tract, concomitant with reduced replication in other cell types, has been observed before [27]. The advantages of the omicron variant are therefore rather coupled to immune evasion, demonstrated by its high resistance to neutralizing antibodies [28,29,30,31]. By using engineered viruses with the same genetic backbone, we have now directly linked the differences in replication kinetics to the influence of the Spike protein. Moreover, our reporter virus system includes a biological safety measure by the deletion of the ORF6 of SARS-CoV-2, which impairs in vivo infection capability [32,33].

As demonstrated in the Figure 5 and Figure 6, the analyzed variants exhibit differences regarding morphology of infected cells, plaque size, and cytotoxicity. Thus, these factors connected to pathogenicity are altered by the respective Spike protein. Additionally, we generated a tool to analyze the impact of different Spike proteins on the syncytia formation. Rajah et al. [34]‚ it was described enhanced syncytia formation for the alpha, beta, and delta variants in comparison to the wild type. Moreover, they identified P681H and D1118H mutations in the Spike protein to facilitate syncytia formation. Furthermore, Saito et al. [35] described the P681R mutation present in most delta-like variants as a potent facilitator of syncytia formation. Consistently, delta and delta70, which contain the P681R mutation, showed the highest capability of syncytia formation, while delta49, without a mutation at amino acid 681, showed a reduced extent of cell−cell fusion (Figure 5a). The reduced syncytia formation in omicron, albeit containing a P681H mutation, might be due to reduced Spike protein cleavage observed for this lineage [26]. In summary, we demonstrate that the differences in infection dynamics are mainly connected to the Spike protein.

A further noticeable advantage of our reporter system is the competitive assay, simultaneously comparing the infection and replication capability of different viruses (Figure 7). By using differently labeled viruses we were able to show that most, but not all, variants are able to outcompete the wildtype virus in vitro. This allows a first, preliminary assessment of the capacity of emerging variants. Additionally, this system enables the analysis of different specific mutations introduced in a reporter virus and their impact on viral infection.

## 4. Material and Methods

### 4.1. Cell Culture

CaCo-2 cells (kindly provided by Konstantin Sparrer, University Hospital Ulm, Germany) were cultivated in DMEM (Dulbecco’s Modified Eagle Medium DMEM; 11500516, Thermo Fisher Scientific, Waltham, MA, USA) supplemented with 10% heat-inactivated fetal bovine serum (FBS-12A; Capricorn Scientific, Ebsdorfergrund, Germany), 2 mM Gluta-MAX™ (35050061, Thermo Fisher Scientific), 25 mM HEPES (15630080, Thermo Fisher Scientific), 1× MEM Non-Essential Amino Acids Solution (11140050, Thermo Fisher Scientific), and 50 µg/mL gentamycin (1405-41-0, Serva Electrophoresis, Heidelberg, Germany) and passaged every 2–3 days depending on confluence.

HEK293T T7/N cells (previously described in [19]) were cultivated in DMEM (Dulbecco’s Modified Eagle Medium DMEM; 11500516, Thermo Fisher Scientific, Waltham, MA, USA) supplemented with 10% heat-inactivated FBS, 2 mM Gluta-MAX™, 25 mM HEPES, 5 µg/mL blasticidin (asnt-bl-1, InvivoGen, San Diego, CA, USA), and 2 µg/mL puromycin (SC-1080713, Santa Cruz Biotechnology, Dallas, TX, USA).

All cells were incubated at 37 °C, 5% CO_2_, and 80% relative humidity.

### 4.2. Bacmid Construction

The backbone for Spike insertion was constructed using Red Recombination as described in [19]. In brief, a kanamycin resistance cassette (KanS) was introduced in the pBeloCoV-Vector with the SARS-CoV-2 B.1 genome in which ORF6 had been replaced by EYFP. The KanS cassette was inserted instead of the Spike gene using Lambda Red Recombination (*E. coli* strain GS1783) and the flanking regions of the kanamycin cassette contained AscI restriction sites.

The respective Spike template cDNA was obtained from residual diagnostic patient samples. Extracted viral RNA was reverse transcribed using LunaScript^®^ RT Supermix (E3010L, New England Biolabs) according to the manufacturer’s protocol. Amplification was achieved using PCR with Q5 DNA polymerase (New England Biolabs, Frankfurt/Main, Germany) and primer pairs containing restriction sites of AscI.

The backbone was digested using AscI (NEB) to remove the kanamycin cassette and Spike amplicon DNA was assembled using the NEBuilder HiFi DNA Assembly Master Mix (E5520, NEB). Constructs were electroporated in *E. coli* GS1783 (1800 V, 25 µF, 200 Ω, 1 mm cuvette).

### 4.3. Infection of CaCo-2 Cells

CaCo-2 cells were seeded 24 h prior to infection. For 96-well cell culture plates 2.5 × 10^4^ cells were seeded, for 12-well cell culture plates the number was 2 × 10^5^. Viruses were extensively vortexed and the respective amount was prepared in 100 µL DMEM supplemented with 5% heat-inactivated FBS, 2 mM Gluta-MAX™, 25 mM HEPES, 1× MEM Non-Essential Amino Acids Solution, and 50 µg/mL gentamycin for 96-well and 1 mL for 12-well. In the case of the 96-well plates the virus solution was pipetted on the original cell culture medium, whereas for the 12-well plates the original cell culture medium was replaced by infection medium. Infected cells were cultivated at 37 °C, 5% CO_2_, and a relative humidity of 80%.

### 4.4. RT-qPCR

Viral supernatant was digested with proteinase K (final concentration of 0.136 mg/mL; PCR-grade, 3115828001, Sigma-Aldrich) for 1 h at 56 °C, followed by inactivation of the enzyme at 95 °C. RT-qPCR was performed using Luna^®^ Universal Probe One-Step RT-qPCR Kit (E3006, NEB). Probes and primers were designed recognizing a fragment of the RNA-dependent RNA-polymerase (RdRp) [19]. Probes were 5′-labeled with VIC (2′-chloro-7′phenyl-1,4-dichloro-6-carboxy-fluorescein). The PCR reaction was performed and measured in an Applied Biosystems 7500 Real-Time PCR system (Applied Biosystems, Waltham, MA, USA).

### 4.5. Virus Reconstitution

To reconstitute active viruses from bacmids, HEK293T cells stably expressing T7 polymerase and viral N protein were seeded in T25 cell culture flasks. After 24 h confluent cells were transfected using 10 µL of purified bacmid and 5 µL of GenJet™ Reagent (II) (SL100489, SignaGen^®^ Laboratories, Frederick, MD, USA) according to the manufacturer’s protocol. After 3–4 days the supernatant (P0) was transferred to confluent CaCo-2 cells in T25 cell culture flasks and replaced by fresh DMEM containing 5% fetal bovine serum, Glutamax, HEPES, and non-essential amino acids after 4–6 h. Fluorescent signal was microscopically controlled after 4–5 days and supernatant was transferred 1:50 to CaCo-2 cells in T75 cell culture flasks. Active viruses were harvested and sterile filtered after detection of strong fluorescence signal or cytopathic effect. Viral titers were determined using endpoint titration (TCID_50_). 

### 4.6. Plaque Assay

CaCo-2 cells were seeded in 12-well cell culture plates 24 h before infection. Viruses were diluted to 50 infectious units in 1 mL of DMEM and cell culture medium was replaced by virus containing medium. The medium was removed 2 h post infection and 1 mL of overlay medium was added (1.6 × 2.4% Avicel RC/CLTM (RC-591), 1× MEM, 1% HEPES (1 M), 1% Glutamax (100×), 10% fetal bovine serum, 2% NaHCO_3_ (1 M)). After 2 days, overlay medium was removed and cells were washed and fixed using 4% paraformaldehyde. Plaques were imaged using an INTAS fluorescence imaging system (Intas Science Imaging, Göttingen, Germany). Plaque area was calculated utilizing free ImageJ2 software, version 1.54d (Wayne Rasband, public domain).

### 4.7. Co-Infection Assay

CaCo-2 cells were seeded in 24-well plates 24 h prior to infection with a density of 1 × 10^5^ per ml (1 mL per well). Viruses were prepared as described in “Infection of CaCo-2 cells” at an MOI = 0.005. After 48 h, supernatants were transferred 1:2000 on freshly seeded CaCo-2 cells in 24-well plates and cells were fixed using 4% paraformaldehyde. Infection rates were determined using the ImageXpress Pico High content Screening device (Molecular Devices, San José, CA, USA) with the Cell reporter express software 2.9.3 in the three channel analysis mode.

### 4.8. Next Generation Sequencing

Viral stocks were centrifuged at 20,000× *g* and 4 °C for 2 h. The supernatant was removed and the pellet was resuspended in Roche buffer. After isolation of viral genomic RNA in a Qiagen EZ-1 instrument, libraries were constructed using a NEBNext ARTIC SARS-CoV-2 FS Library Prep Kit (E7658, NEB). Paired-end sequencing was performed with the MiSeq reagent kit v3 (150 cycles) on a MiSeq™ Instrument (Illumina, San Diego, CA, USA). Sequences were analyzed utilizing CLC Genomics Workbench 22 (Qiagen Aarhus A/S, Aarhus, Denmark). 

### 4.9. Biosafety

All work with bacmid-generated SARS-CoV-2 virus was performed in the registered BSL3 facility at the Institute of Virology at University Hospital of the Friedrich-Alexander-University Erlangen-Nuremberg (Az. 821-8760.00-23/90; Az. 821-8791.2.12; Az. 821-8791.2.13). Generation of recombinant SARS-CoV-2 by the bacmid method was approved by the Central Committee for Biological Safety (ZKBS) consulting the German Federal Office of Consumer Protection and Food Safety (BVL) (Az. 45110.2084 and Az. 45110.2220). Permission for the experiments was granted by the regional authorities of Lower Franconia (Az. 55.1-8791.27-28-20, Az. 55.1-8791.27-29-20, and Az. 55.1-8791.27-28-23).

## Figures and Tables

**Figure 1 ijms-24-08156-f001:**
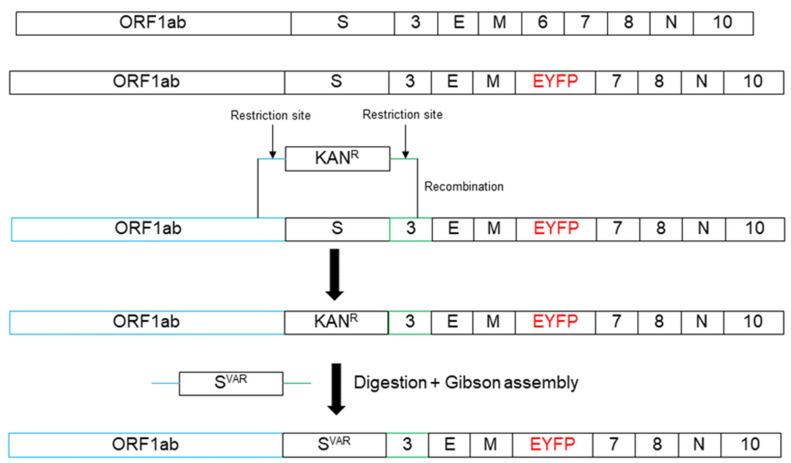
Cloning strategy for SARS-CoV-2 Spike variants. The Spike protein coding sequence is replaced via homologous recombination by a kanamycin resistance cassette flanked by AscI restriction enzyme cleavage sites. In a second recombination step this cassette is replaced by the Spike protein sequence of interest. Furthermore, the recombinant virus expresses an EYFP reporter gene instead of ORF6 for visualization of infected cells and as biological safety measure.

**Figure 2 ijms-24-08156-f002:**
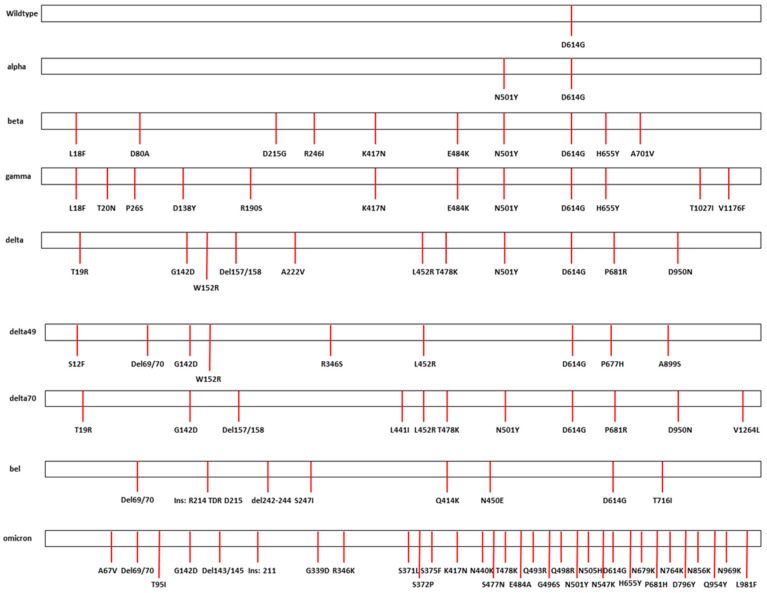
Mutation patterns of analyzed Spike variants. The Spike proteins of the constructed SARS-CoV-2 variants are shown and mutations are visualized by red bars. Wild type = B.1; alpha = B.1.1.7; beta = B.1.351; gamma = B.1.1.28.1; delta = B.1.617.2, bel = B.1.214.2, omicron = B.1.1.529.

**Figure 3 ijms-24-08156-f003:**
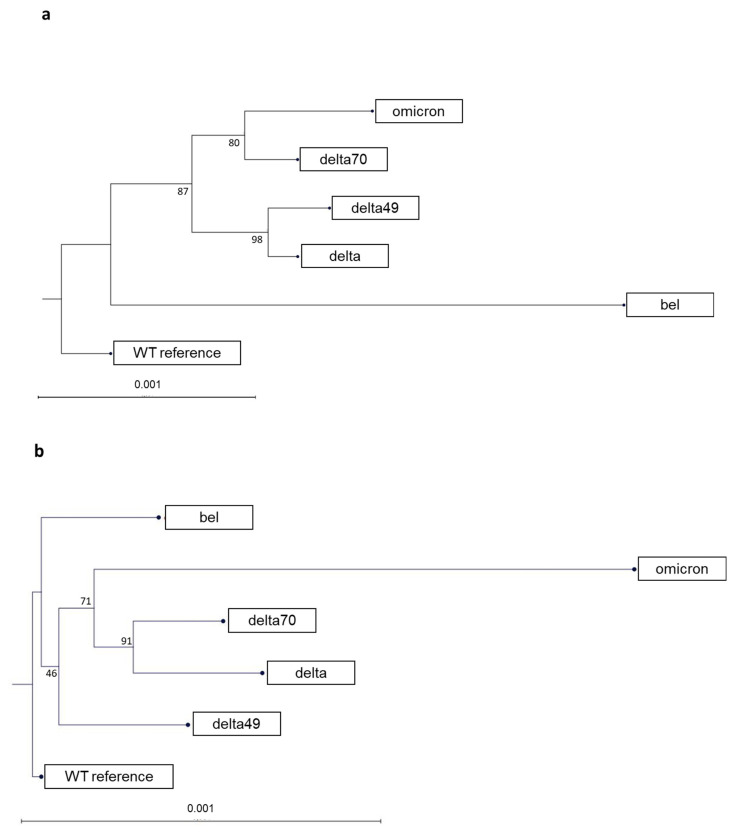
Phylogenetic relationship of the analyzed variants. The trees were generated by aligning either complete genome sequences (**a**) or Spike gene sequences only (**b**) using CLC genomics workbench. The respective alignment based on Jukes−Cantor model was used for maximum likelihood tree generation with 1000 repeats. Scale bars indicate the phylogenetic distance of number of substitutions/changes per nucleotide.

**Figure 4 ijms-24-08156-f004:**
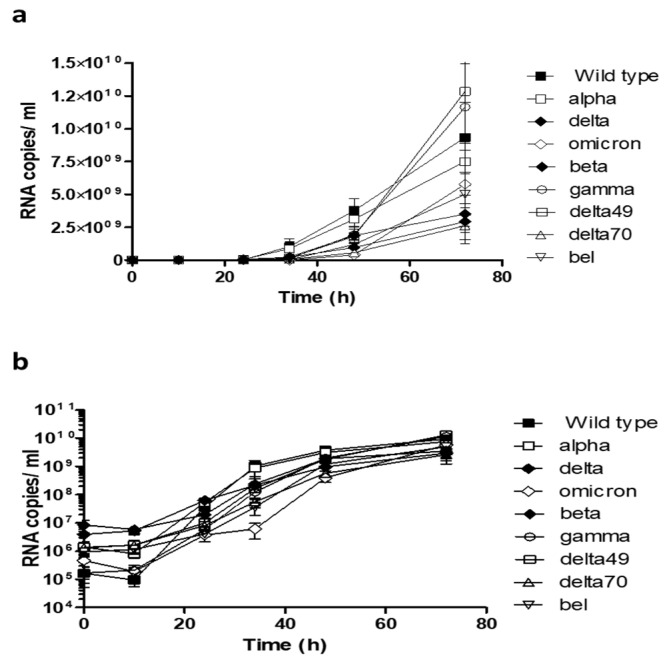
Replication characteristics of SARS-CoV-2 Spike variants. CaCo-2 cells were infected at a MOI of 0.005 and samples were taken at the indicated time points. The viral load was determined by RT-qPCR targeting the RNA-dependent RNA-Polymerase. The mean of three independent experiments in (**a**) linear scale and (**b**) logarithmic scale is shown. Error bars indicate standard deviation.

**Figure 5 ijms-24-08156-f005:**
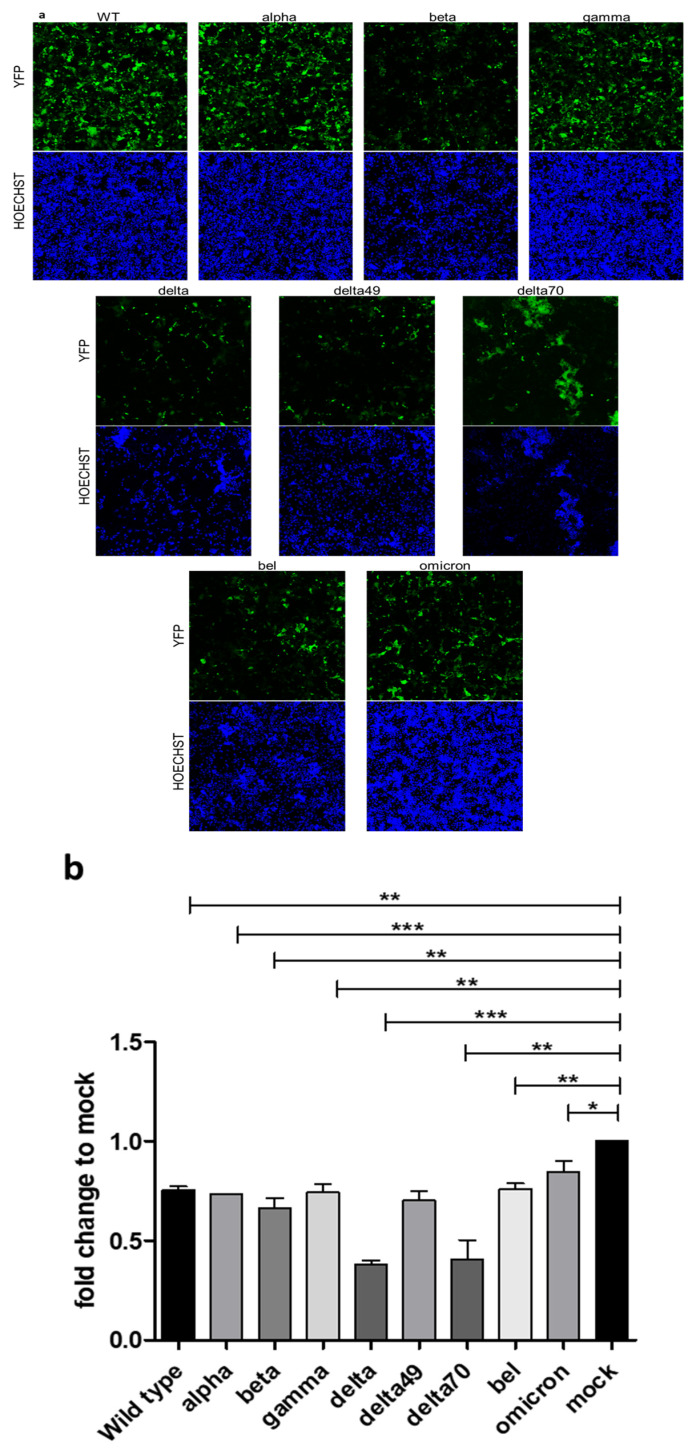
Morphology and cell survival of CaCo-2 cells infected with the SARS-CoV-2 Spike variants. CaCo-2 cells were infected with SARS-CoV-2 Spike variants at a MOI of 0.005 and fixed after 48 h. Subsequently, fluorescent measurement was performed. (**a**) Fluorescent images of the different variants. (**b**) Cell survival in fold change relative to mock based on HOECHST measurement in a Victor X4 plate reader. The mean of three independent experiments is shown. Error bars indicate standard deviation. Significance: * = *p* < 0.05, ** = *p* < 0.01, *** = *p* < 0.001.

**Figure 6 ijms-24-08156-f006:**
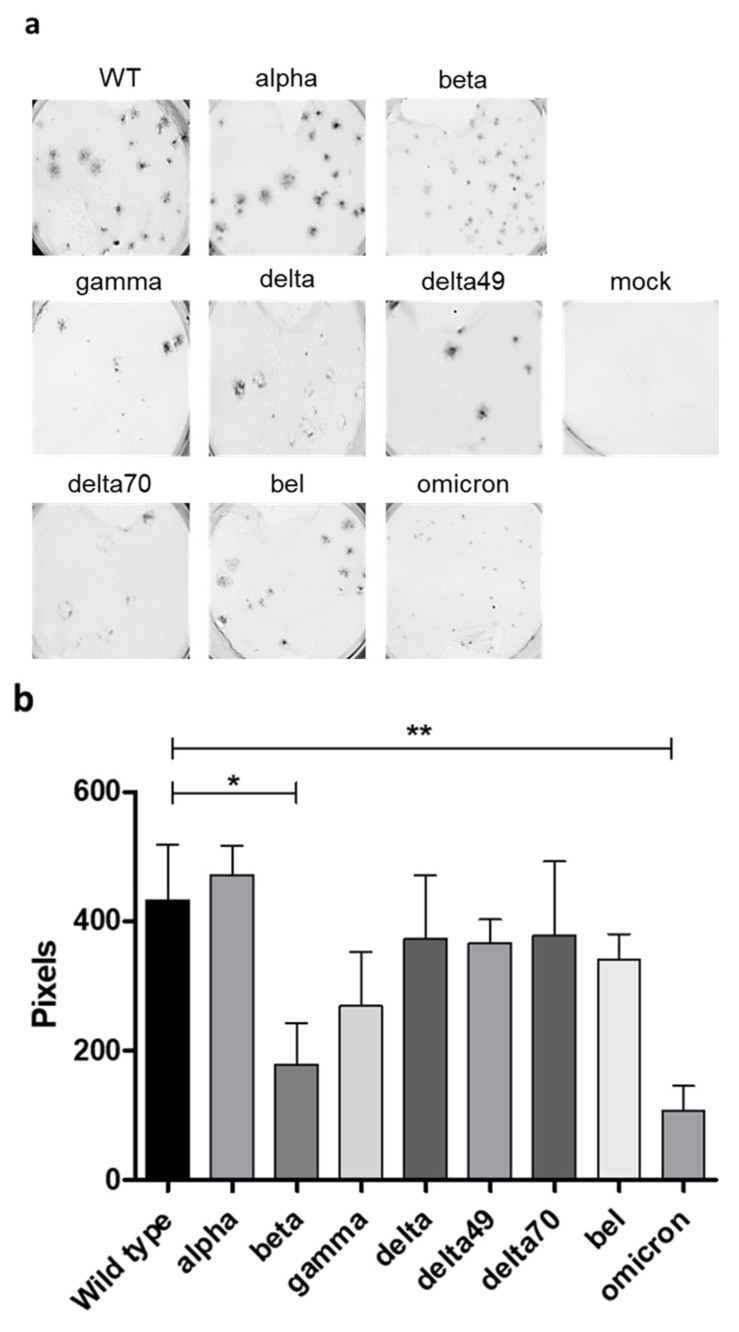
Plaque morphology of SARS-CoV-2 Spike variants. CaCo-2 cells were infected with 50 infectious viral particles per well and covered with overlay medium. After 48 h, cells were fixed and fluorescent images were taken. Images (**a**) and calculated plaque areas (**b**) are shown. Three independent experiments were performed. Error bars indicate standard deviation. * = *p* < 0.05, ** = *p* < 0.01, not significant if not indicated otherwise.

**Figure 7 ijms-24-08156-f007:**
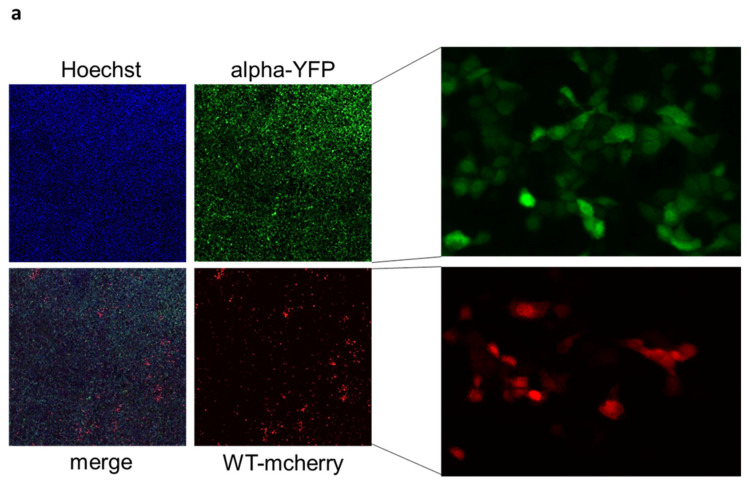
Competitive growth of SARS-CoV-2 Spike variants and wild type. (**a**) Example of fluorescent image of co-infection. (**b**) CaCo-2 cells were co-infected with wild type SARS-CoV-2 (mCherry) and one Spike variant (EYFP) at an MOI of 0.005 each. Supernatants were passaged every 48 h and cells fixed for fluorescent imaging. The mean of three independent experiments is depicted.

## Data Availability

The data presented in this study are available on request from the corresponding author.

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
