# Peer review of "Analysis of SARS-CoV-2 Spike Protein Variants with Recombinant Reporter Viruses Created from a Bacmid System"

_ijms, 2023, doi:10.3390/ijms24098156_

Round 1

Reviewer 1 Report

The authors presented an interesting work which includes a new method based on previously described SARS-CoV-2 bacmid system (developed by the same group) to assess the influence of SARS-CoV-2 S-spike protein (and potentially other proteins) on the virus infection dynamics.  The authour have compared the replication properties and infection parameters of a series of  recombinant reporter viruses possessing Spike genes from different VoCs as well as some more exotic Spike genes from viruses causing small local transmission events in the area of Erlangen (Bavaria, Germany). One of the important advantages of the represented system is that it allows to analyze the "natural" genes taken from the patients' samples. Since the same virus background is used in all compared samples one can follow the influence of the exact mutations in the S-spike protein.

The work is very accurately carried out and represented. The discussion is relevant and conclusions are clear.

I have no specific suggestions to improve the paper.

Hopefully, some small missprints (very few, e.g. line 168, differenz instead of different) will be corrected during the copy editing stage.

Author Response

Reviewer 1: We thank the reviewer for the positive comments and have checked the manuscript extensively for spelling errors and other incorrectness. We have also reformulated a few complicated long sentences, as suggested.

In addition, we have appended a paragraph with information regarding Biosafety and respective permissions by the regulatory authorities. (line 332ff)

Reviewer 2 Report

 Major

1.      Are there plans to dissect which specific variant mutations account for the increased pathogenicity of the delta variants using the bacmid system? I believe that inclusion of this data would have significantly increased the impact of the manuscript.

2.      In vivo testing would have been insightful and provided clarity into pathological changes dependent on spike mutations. Will this be done in the future?

3.      It is interesting to note that the replication kinetics for each variant is relatively similar, but infectivity and plaque size diverge. Obviously, receptor binding mutations play a role here, but variant mutations appear to influence cell survival. Please provide more insight into how the spike mutations are impacting pathogenicity of each variant. The authors cite several other works, but do not provide details. For example, the authors cite Rajah et al (2021) when discussing syncytia formation, but do not relate how these mutations are altering the infectivity measured here. Providing a bit more background/explanation may help to clarify the results and inform the reader.

4.      In Figure 5, the fluorescent YFP analysis of the beta virus appears equivalent to the delta and delta49 strains, but beta has a similar cell survival rate as wild type. Please provide insight to this seeming discordance.

The structure of some sentences is difficult to understand. It may be helpful to edit some of the language.

Author Response

Reviewer 2: We thank the reviewer for the positive comments and have integrated the feedback:

  1. Yes, we are planning to analyze specific mutations of delta mutations in the future. We have added a respective comment in the manuscript. (line 206ff)
  2. In the same line, we plan to do in vivo Our BSL3 facility is currently upgraded and equipped with IVC incubators for mouse experiments; permission for animal experiments by authorities has been applied for and is pending. We have added a brief sentence at the respective part of the manuscript. (line 206ff)
  3. We have added details to the description of syncytia formation. (line 226ff)
  4. The images were all taken with the Pico HCS instrument with the same capture settings that were set to minimize overexposure and allow quantitative analysis over a 10-bit linear range (4096 grayscales). The HOECHST staining of the beta Spike variant thereby appears weaker in the imaging than for the others that had more bright signals of dying cells. Therefore, the image appears skewed in that cells that are present may not be seen in the fluorescent image. However, these cells could be measured by the plate reader.

We have also reformulated a few complicated long sentences, as suggested.

In addition, we have appended a paragraph with information regarding Biosafety and respective permissions by the regulatory authorities. (line 332ff)